# Optimization and Generalization of Shallow Neural Networks with Quadratic Activation Functions

**Stefano Sarao Mannelli**[†‡∗], **Eric Vanden-Eijnden**[‡], and **Lenka Zdeborová**[§]

## Abstract

We study the dynamics of optimization and the generalization properties of one-hidden layer neural networks with quadratic activation function in the over-parametrized regime where the layer width $m$ is larger than the input dimension $d$. We consider a teacher-student scenario where the teacher has the same structure as the student with a hidden layer of smaller width $m^* \leq m$. We describe how the empirical loss landscape is affected by the number $n$ of data samples and the width $m^*$ of the teacher network. In particular we determine how the probability that there be no spurious minima on the empirical loss depends on $n$, $d$, and $m^*$, thereby establishing conditions under which the neural network can in principle recover the teacher. We also show that under the same conditions gradient descent dynamics on the empirical loss converges and leads to small generalization error, i.e. it enables recovery in practice. Finally we characterize the time-convergence rate of gradient descent in the limit of a large number of samples. These results are confirmed by numerical experiments.

## 1 Introduction

Neural networks are a key component of the machine learning toolbox. Still the reasons behind their success remain mysterious from a theoretical prospective. While sufficiently large neural networks can in principle represent a large class of functions, we do not yet understand under what conditions their parameters can be adjusted in an algorithmically tractable way for that purpose. For example, under worst case assumptions, some functions cannot be tractably learned with neural networks [1, 2]. We also know that there exist settings with adversarial initializations where neural networks fail in generalization to new samples, while the same setting from random initial conditions succeeds [3]. And yet, in many practical settings, neural networks are trained successfully even with simple local algorithm such as gradient descent (GD) or stochastic gradient descent (SGD).

The problem of learning the parameters of a neural network is two-fold. First, we want that their training on a set of data via minimization of a suitable loss function succeed in finding a set of parameters for which the value of the loss is close to its global minimum. Second, and more importantly, we want that such a set of parameters also generalizes well to unseen data. Theoretical guarantees have been obtained in many settings by a geometrical analysis of the loss showing that only global minima are present, see e.g. [4, 5]. In particular it has been shown that network over-parametrization can be beneficial and lead to landscapes without spurious minima in which GD or SGD converge [6–10]. However, over-parametrized neural networks successfully optimized on a training set do not necessarily generalize well – for example neural networks can achieve zero errors

∗ Work done while visiting at Courant Institute.

in training without learning any rule [11]. It is therefore important to understand when zero training loss implies good generalization.

It is know empirically that deep neural networks can learn functions that can be represented with a much smaller (sometimes even shallow) neural network [12–14], but that learning the smaller network without first learning the larger one is computationally harder [6]. Our work provides a theoretical justification for this empirical observation by providing an explicit and rigorously analyzable case where this happens.

**Main contributions:** In this work we investigate the issues of training and generalization in the context of a teacher-student set-up. We assume that both the teacher and the student are one-hidden layer neural network with quadratic activation function and quadratic loss. We focus on the over-parametrized or over-realizable case where the hidden layer of the teacher $m^*$ is smaller than that of the student $m$. We assume that the hidden layer of the student $m$ is larger than the dimensionality $d$, $m > d$, in that case:

- We show that the value of the empirical loss is zero on all of its minimizers, but that the set of minimizers does not reduce to the singleton containing only the teacher network in general.

- We derive a critical value $\alpha_c = m^* + 1$ of the number of samples $n$ per dimension $d$ above which the set of minimizers of the empirical loss has a positive probability to reduce to the singleton containing only the teacher network in the limit $n, d \to \infty$ with $n/d \geq \alpha_c$— i.e. we derive a sample complexity threshold above which the minimizer can have good generalization properties. The formula is proven for a teacher with a single hidden unit $m^* = 1$ (a.k.a. phase retrieval).

- We study gradient descent flow on the empirical loss starting from random initialization and show that it converges to a network that can achieve perfect generalization above this sample complexity threshold $\alpha_c$.

- We quantify the nonasymptotic convergence rate of gradient descent in the limit of large number of samples and show that the loss is bounded from above at all times by $C_1/(1+C_2 t)$ for some constants $C_1, C_2 > 0$. We also evaluate the asymptotic convergence rate and identify two different regimes according to the input dimension and the number of hidden units, showing that in one case the loss converges as $O(t^{-2})$ as $t \to \infty$ while in the second case it converges exponentially.

- We show how the string method can be used to probe the empirical loss landscape and find minimum energy paths on this landscape connecting the initial weights of the student to those of the teacher, possibly going through flat portion or above energy barrier. This allows one to probe features of this landscape not accessible by standard GD.

In Sec. 2 we formally define the problem and derive some key properties that we use in the rest of the paper. In Sec. 3 we analyze the training and the generalization losses from the geometrical prospective, and derive the formula for the sample complexity threshold. In Sec. 4 we show that gradient descent flow can find good minima for datasets above this sample complexity threshold, and we characterize its convergence rate. In Sec. 6 we present our results using the string method to probe the loss landscape. Finally in the appendix we give the proofs and some additional numerical results.

**Related works:** One-hidden layer neural networks with quadratic activation functions in the over-parametrized regime were considered in a range of previous works [8,9,15–17]. Notably it was shown that all local minima are global when the number of hidden units $m$ is larger than the dimension $d$ and that gradient descent finds the global optimum [8, 15, 16], and also when the number of hidden units $m > \sqrt{2n}$ with $n$ being the number of samples [15, 17]. Most of these results were established for arbitrary training data of input/output pairs, but consequently these works did not establish condition under which the minimizers reached by the gradient descent have good generalization properties. Indeed, it is intuitive that over-parametrization renders the optimization problem simpler, but it is rather non-intuitive that it does not destroy good generalization properties. In [15], under the assumption that the input data is Gaussian i.i.d., a $O(1/\sqrt{n})$ generalization rate was established. However the generalization properties of neural networks with number of samples comparable to the dimensionality is mostly left open.

Much tighter (Bayes-optimal) generalization properties of neural networks were established for data generated by the teacher-student model, for the generalized linear models in [18], and for one hidden layer much smaller than the dimension in [19]. However, these results were only shown to be achievable with approximate message passing algorithms and the performance of gradient-descent algorithm was not analyzed. Also studying over-parametrization with analogous tightness of generalization results is an open problem and has been achieved only for the one-pass stochastic gradient descent [20].

A notable special case of our setting is when the teacher has only one hidden unit, in which case the teacher network is equivalent to the phase retrieval problem with random sensing matrix [21]. For this case the performance of message passing algorithms is well understood and requires a number of samples linearly proportional to the dimension, $n > 1.13d$ in the high-dimensional regime for perfect generalization [18]. For randomly initialized gradient descent the best existing rigorous result for the phase retrieval requires $d\mathrm{poly}(\log d)$ number of samples [22]. The performance of the gradient-descent in the phase retrieval problem is studied in detail in a concurrent work [23], showing numerically that without overparametrization randomly initialized gradient descent needs at least $n \approx 7d$ samples to find perfect generalization. In the present work we show that overparametrized neural networks are able to solve the phase retrieval problem with $n > 2d$ samples in the high-dimensional limit. This improves upon [22] and falls close to the performance of the approximate message passing algorithm that is conjectured optimal among polynomial ones [18]. But most interesting is the comparison between our results for the phase retrieval obtained by overparametrized neural networks $\alpha_c = 2$, and the results from [23] who show that without overparametrized considerably larger $\alpha$ is needed for gradient descent to succeed to learn the same function. This comparison provides a theoretical justification for how overparametrization helps gradient descent to find good generalization properties with fewer samples. We stress that the same property would not apply to the message passing algorithms. We could speculate that more of the properties of overparametrization observed in deep learning are limited to the gradient-descent-based algorithms and would not hold for other algorithmic classes.

Closely related to our work is Ref. [24] in which the authors consider the same teacher-student problem as we do. The main difference is that they only consider teachers that have more hidden units than the input dimension, $m^* \geq d$, while we consider arbitrary $m^*$. As we show below the regime where $m^* < d$ turns out to be interesting as it affects nontrivially the critical number of samples $n_c$ needed for recovery and leads to a more complex scenario in which $n_c$ depends also on $m^*$—in particular taking $m^* < d$ allows for recovery below the threshold $d(d+1)/2$, which is one of our main results.

## 2  Problem formulation

Consider a teacher-student scenario where a teacher network generates the dataset, and a student network aims at learning the function of the teacher. The teacher has weights $\boldsymbol{w}_i^* \in \mathbb{R}^d$, with $i = 1, \ldots, m^*$. We will keep the teacher weights generic in most of the paper and will specify them when needed, in particular for the simulations where we consider two specific teachers: one with $\{\boldsymbol{w}_i^*\}_{i \leq m^*}$ i.i.d. Gaussian with covariance identity, and one with $\{\boldsymbol{w}_i^*\}_{i \leq m^*}$ orthonormal.

The student's weights are $\boldsymbol{w}_j \in \mathbb{R}^d$, with $j = 1, \ldots, m$ and $m \geq d$. Given an input $\boldsymbol{x} \in \mathbb{R}^d$, teacher's and student's outputs are respectively

$$f_*(\boldsymbol{x}) = \frac{1}{m^*} \sum_{i=1}^{m^*} |\boldsymbol{x} \cdot \boldsymbol{w}_i^*|^2, \qquad \text{and} \qquad f(\boldsymbol{x}) = \frac{1}{m} \sum_{j=1}^{m} |\boldsymbol{x} \cdot \boldsymbol{w}_j|^2, \tag{1}$$

where we fixed the second layer of weights to $1/m^*$ and $1/m$, respectively. The teacher produces $n$ outputs $y_k = f_*(\boldsymbol{x}_k)$ from random i.i.d. Gaussian samples $\boldsymbol{x}_k \sim \nu = \mathcal{N}(0, I_d)$, $k = 1, \ldots, n$. Given this dataset, we define the empirical loss

$$L_n(\boldsymbol{w}_1, \ldots, \boldsymbol{w}_m) = \frac{1}{4} \mathbb{E}_{\nu_n} \left| \frac{1}{m^*} \sum_{i=1}^{m^*} |\boldsymbol{x} \cdot \boldsymbol{w}_i^*|^2 - \frac{1}{m} \sum_{j=1}^{m} |\boldsymbol{x} \cdot \boldsymbol{w}_j|^2 \right|^2 \tag{2}$$

where $\mathbb{E}_{\nu_n}$ denotes expectation with respect to the empirical measure $\nu_n = n^{-1} \sum_{k=1}^{n} \delta_{\boldsymbol{x}_k}$. As usual, the population loss is obtained by taking the expectation of (2) with respect to $\nu$.

The student minimizes the empirical loss (2) using gradient descent, $\dot{\boldsymbol{w}}_i(t) = -m\partial_{\boldsymbol{w}_i} L_n$. Explicitly

$$\dot{\boldsymbol{w}}_i(t) = \mathbb{E}_{\nu_n}\left[\operatorname{tr}\left(X(A^* - A(t))\right)X\boldsymbol{w}_i(t)\right]. \tag{3}$$

where we introduced the following $d \times d$ matrices

$$A(t) = \frac{1}{m}\sum_{i=1}^{m}\boldsymbol{w}_i(t)\boldsymbol{w}_i^T(t), \quad A^* = \frac{1}{m^*}\sum_{i=1}^{m^*}\boldsymbol{w}_i^*(\boldsymbol{w}_i^*)^T, \quad X = \boldsymbol{x}\boldsymbol{x}^T. \tag{4}$$

We can now see that a closed equation for $A(t)$ can be derived from (3), and this new equation reduces the effective number of weights from $O(dn)$ to $O(d^2)$ without affecting neither the dynamics nor the other properties of the teacher and student since $f_*(\boldsymbol{x}) = \operatorname{tr}(XA^*)$ and $f(\boldsymbol{x}) = \operatorname{tr}(XA)$:

**Lemma 2.1.** *The GD flow* (3) *of the weights* $\{\boldsymbol{w}_i\}_{i\leq m}$ *on the empirical loss induces the following evolution equation for $A(t)$:*

$$\dot{A} = -A\nabla E_n(A) - \nabla E_n(A)A = \mathbb{E}_{\nu_n}[\operatorname{tr}\left(X(A^* - A)\right)(AX + XA)], \tag{5}$$

*where $\nabla$ denotes gradient with respect to $A$ and $E_n(A)$ is twice the empirical loss* (2) *rewritten in terms of $A$:*

$$E_n(A) = \frac{1}{2}\mathbb{E}_{\nu_n}\left|\operatorname{tr}\left(X(A - A^*)\right)\right|^2. \tag{6}$$

It is also possible to write the equivalent of this lemma for the population loss:

**Lemma 2.2.** *The GD flow of the weights* $\{\boldsymbol{w}_i\}_{i\leq m}$ *on the population loss reads*

$$\dot{\boldsymbol{w}}_i(t) = \operatorname{tr}(A^* - A(t))\boldsymbol{w}_i(t) + 2(A^* - A(t))\boldsymbol{w}_i(t). \tag{7}$$

*and it induces the following evolution equation for $A(t)$:*

$$\dot{A} = -A\nabla E(A) - \nabla E(A)A = 2\left[(\operatorname{tr}(A^* - A))A + (A^* - A)A + A(A^* - A)\right]. \tag{8}$$

*where $E(A)$ is twice the population loss written in terms of $A$:*

$$E(A) = \operatorname{tr}\left((A - A^*)^2\right) + \frac{1}{2}\left(\operatorname{tr}(A - A^*)\right)^2. \tag{9}$$

Expression (9) for the population loss was already given in [24]. Lemmas 2.1 and 2.2 are proven in Appendices A.1 and A.2, respectively. In Appendix A.3 we also show that (5) and (8) are the continuous limit of proximal schemes on $E_n$ and $E$, respectively, relative to a specific Bergman divergence.

## 3   Geometrical Considerations and Sample Complexity Threshold

The empirical loss $E_n(A)$ is quadratic, hence convex, with minimum zero. In addition $A = A^*$ is a minimizer since $E_n(A^*) = 0$. The main question we want to address next is when is this minimizer unique.

Since the trace is a scalar product in the vector space of $d \times d$ matrices in which symmetric matrices form a $d(d+1)/2$ dimensional subspace, the empirical loss $E_n(A)$ will be strictly convex in this subspace *iff* we span it using $d(d+1)/2$ linearly independent $X_k = \boldsymbol{x}_k\boldsymbol{x}_k^T$ [24]. Yet, if we restrict considerations to matrices $A$ that are also positive semidefinite, we need less data to guarantee that $A = A^*$ is the unique minimizer of $E_n(A)$, at least in some probabilistic sense:

**Theorem 3.1** (Single unit teacher). *Consider a teacher with $m^* = 1$ and a student with $m \geq d$ hidden units respectively, so that $A^*$ has rank 1 and $A$ has full rank. Given a data set $\{\boldsymbol{x}_k\}_{k=1}^n$ with each $\boldsymbol{x}_k \in \mathbb{R}^d$ drawn independently from a standard Gaussian, denote by $\mathcal{M}_{n,d}$ the set of minimizer of the empirical loss constructed with $\{\boldsymbol{x}_k\}_{k=1}^n$ over symmetric positive semidefinite matrices $A$, i.e.*

$$\mathcal{M}_{n,d} = \left\{A = A^T, \text{ positive semidefinite such that } E_n(A) = 0\right\}. \tag{10}$$

*Set $n = \lfloor\alpha d\rfloor$ for $\alpha \geq 1$ and let $d \to \infty$. Then*

$$\lim_{d\to\infty}\mathbb{P}\left(\mathcal{M}_{\lfloor\alpha d\rfloor,d} \neq \{A^*\}\right) = 1 \qquad \text{if } \alpha \in [0, 2] \tag{11}$$

*whereas*

$$\lim_{d\to\infty}\mathbb{P}\left(\mathcal{M}_{\lfloor\alpha d\rfloor,d} = \{A^*\}\right) > 0 \qquad \text{if } \alpha \in (2, \infty). \tag{12}$$

In words, this theorem says that it exists a threshold value $\alpha_c = 2$ such that for any $n > n_c = \lfloor \alpha d \rfloor$ there is a finite probability that the empirical loss landscape trivializes and all spurious minima disappear in the limit as $d \to \infty$. For $n \le n_c$ however, this is not the case and spurious minima exist with probability 1 in the limit. Therefore, the chance to learn $A^*$ by minimizing the empirical loss from a random initial condition is zero if $\alpha \in [0, 2)$ but it becomes positive if $\alpha > 2$. The proof of Theorem 3.1 is presented in Appendix A. This proof shows that we can account for the constraint that $A$ be positive definite by making a connection with the classic problem of the number of extremal rays of proper convex polyhedral cones generated by a set of random vectors in general position. Interestingly, this proof also gives a criterion on the data set $\{ \boldsymbol{x}_k \}_{k=1}^n$ that guarantees that the only minimizer of the empirical loss be $A^*$: it suffices to check that the proper convex polyhedral cones constructed with the data vectors have a number of extremal rays that is less than $n$.

**Heuristic extension for arbitrary $m^*$.** The result of Theorem 3.1 can also be understood via a heuristic algebraic argument that has the advantage that it applies to arbitrary $m^*$. The idea, elaborated upon in Appendix A.5, is to count the number of constraints needed to ensure that the only minimum of the empirical loss is $A = A^*$, taking into account that (i) $A$ has full rank and $A^*$ has rank $m^*$ and (ii) both $A$ and $A^*$ are positive semidefinite and symmetric, so that the number of negative eigenvalues of $A - A^*$ can at most be $m^*$. If we use a block representation of $A - A^*$ in which we diagonalize the block that contains the direction associated with the eigenvectors of $A - A^*$ with nonnegative eigenvalues, and simply count the number of nonzero entries in the resulting matrix (accounting for its symmetry), for $m^* < d$ we arrive at

$$n_c = d(m^* + 1) - \tfrac{1}{2} m^* (m^* + 1) \tag{13}$$

while for $m^* \ge d$ we recover the result $n = d(d+1)/2$ already found in [9, 24]. Setting $n_c = \alpha_c d$ and sending $d \to \infty$, this gives the sample complexity threshold

$$\alpha_c = (m^* + 1) \tag{14}$$

which, for $m^* = 1$, agrees with the result in Theorem 3.1. The sample complexity threshold is confirmed in Fig. 1 via simulations using gradient descent (GD) on the empirical loss—we explain this figure in Sec. 4 after establishing that the GD dynamics converges.

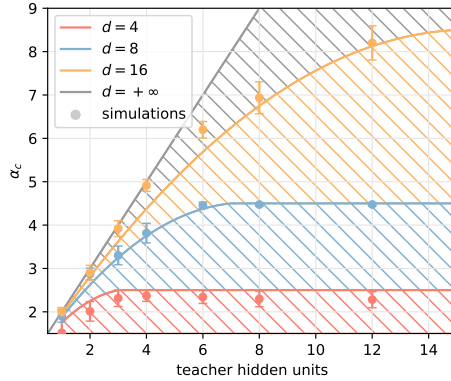

Figure 1: Dynamical phases of the student performance with a teacher having a number of hidden units given on the $x$-axis. The solid lines show the theoretical prediction in (13) for the sample complexity threshold and the points are obtained by extrapolation from simulations with GD. In the simulations we consider a teacher with i.i.d. Gaussian weights and we report other cases in the Appendix.

## 4 Convergence of Gradient Descent on the Empirical Loss

Let us now analyze the performance of gradient descent over the empirical loss. As shown in Appendix A.6, we can prove that:

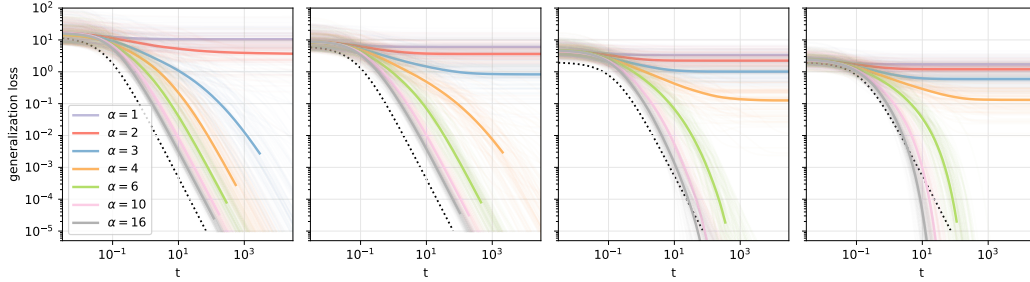

Figure 2: Convergence rates increasing the number of hidden units in the teacher $m^*$. The figures show log-average of 100 simulations with $d = 8$ and from left to right $m^* = 2, 4, 8, 16$, respectively. The individual simulations are shown in transparency. The dotted line is the quadratic decay and serves as reference. The figure shows that, if $\alpha > \alpha_c$ and $m^* > d - 1$ the convergence rate becomes faster than quadratic, and in fact exponential as derived in Sec. 4.

**Theorem 4.1.** *Let $\{w_i(t)\}_{i=1}^m$ be the solution to* (3) *for the initial data $\{w_i(0)\}_{i=1}^m$. Assume that $m \geq d$ and each $w_i(0)$ is drawn independently from a distribution that is absolutely continuous with respect to the Lebesgue measure on $\mathbb{R}^d$. Then*

$$A = \frac{1}{m} \sum_{i=1}^m w_i(t) w_i^T(t) \to A_\infty = \frac{1}{m} \sum_{i=1}^m w_i^\infty (w_i^\infty)^T \quad as \ t \to \infty \tag{15}$$

*and $A_\infty$ is a global minimizer of the empirical loss, i.e.*

$$E_n(A_\infty) = 2L_n(w_1^\infty, \dots, w_n^\infty) = 0. \tag{16}$$

In a nutshell this theorem can be proven using the equivalence between the formulation using the weights with the GD flow in (3) over the loss $L_n$ in (2) and that using $A$ with the evolution equation in (5) and the associated loss $E_n$ in (6). We can invoke the Stable Manifold Theorem [25] to assert that the solution (3) must converge to a local minimum of $L_n$; as soon as $m \geq d$ and $A(0)$ has full rank, this minimum must be a minimum of $E_n$, which means that it must be the global since $E_n$ is convex. Note also that Theorem 4.1 can be generalized to time-discretized version of the GD flow using the results in Ref. [7]

Combined with Theorem 3.1, Theorem 4.1 indicates that, when $m^* = 1$ and $d$ is large, the probability that $A_\infty \neq A^*$ is high when $n/d \geq 2$, whereas the probability that $A_\infty = A^*$ becomes positive for $n/d > 2$. If we generalize this analysis to the case $m^* > 1$ and $d$ large, we expect that GD will recover the teacher only if $n \geq n_c$ with $n_c$ given by (13).

These results are confirmed by numerical simulations in Fig. 1 where we plot $\alpha_c = n_c/d$ as a function of the number of teacher hidden units $m^*$ for different values of $d$. The four colors represent different input dimensions $d = 4, 8, 16, \infty$. We use circles to represent the numerical extrapolation of $\alpha_c$ obtained by several runs of GD flow on different instances of the problem, using the procedure described in Appendix B. Consistent with Theorem 4.1, the extrapolation confirms that GD flow is able match the sample complexity threshold predicted by the theory.

## 5 Convergence Rate of Gradient Descent on the Population Loss

Theorems 4.1 leaves open is the convergence rate of $A(t)$ towards $A_\infty$. This question is hard to answer for GD on the empirical loss, but it can be addressed for GD on the population loss.

**Theorem 5.1.** *Let $\{w_i(t)\}_{i=1}^m$ be the solution to* (7) *for the initial data $\{w_i(0)\}_{i=1}^m$. Assume that $m \geq d$ and each $w_i(0)$ is drawn independently from a distribution that is absolutely continuous with respect to the Lebesgue measure on $\mathbb{R}^d$. Then*

$$A(t) = \frac{1}{m} \sum_{i=1}^m w_i(t) w_i^T(t) \to A^* \quad as \ t \to \infty \tag{17}$$

*and we have the following nonasymptotic bound on the convergence rate of the population loss* (9):

$$\exists C > 0 \quad : \quad \forall t \geq 0 \qquad E(A(t)) \leq \frac{E(A(0))}{1 + 2CE(A(0))t} \tag{18}$$

*In addition $E(A(t))$ decays faster than $1/t$ as $t \to \infty$, i.e. $E(A(t)) = o(1/t)$ and*

$$\lim_{t \to \infty} tE(A(t)) = 0. \tag{19}$$

This theorem is proven in Appendix A.7. The proof uses the convexity of $E(A)$ and deals with the added complexity of the factors $A$ multiplying $\nabla E$ in (8). The argument also uses a stochastic representation formula for $A^{-1}(t)$ given in Lemma A.2 which is interesting in its own right. We stress that (18) holds even when $m^* < d$, i.e. when $A^*$ is rank deficient, which is the difficult case for analysis since the factors $A$ multiplying $\nabla E$ in (8) converge to $A^*$ and hence become only positive semidefinite (as opposed to positive definite) as $t \to \infty$.

Theorem 5.1 holds for arbitrary initial conditions $A(0)$ with full rank. If the initial weights $\boldsymbol{w}_i(0)$ are drawn independent from a standard Gaussian distribution in $\mathbb{R}^d$, we know that $A(0) = m^{-1} \sum_{i=1}^m \boldsymbol{w}_i(0)\boldsymbol{w}_i^T(0) \to \mathrm{Id}$ almost surely as $m \to \infty$ by the Law of Large Numbers. Therefore it makes sense to consider the GD flow (8) on the population loss when $A(0) = \mathrm{Id}$. In that case, we have:

**Theorem 5.2.** *Let $A(t)$ be the solution to (8) for the initial condition $A(0) = \mathrm{Id}$. Denote by $U^*$ an orthogonal matrix whose columns are the eigenvectors of $A^*$, so that $A^* = U^*\Lambda^*(U^*)^T$ with $\Lambda^* = diag(\lambda_1^*, \ldots, \lambda_d^*)$. Let $\Lambda(t) = (U^*)^T A(t) U^*$ so that $\Lambda(0) = \mathrm{Id}$. Then $\Lambda(t)$ remains diagonal during the dynamics and the evolution of its entries is given by*

$$\dot{\lambda}_i = 2\sum_{j=1}^d (\lambda_j^* - \lambda_j)\lambda_i + 4(\lambda_i^* - \lambda_i)\lambda_i, \quad \lambda_i(0) = 1, \qquad i = 1, \ldots, d. \tag{20}$$

*In addition the population loss is given by*

$$E[A(t)] = \sum_{j=1}^d (\lambda_j(t) - \lambda_j^*)^2 + \frac{1}{2}\Big(\sum_{j=1}^d \lambda_j(t) - \lambda_j^*\Big)^2. \tag{21}$$

This theorem is proven in Appendix A.8. The equations in (20) can easily be solved numerically. A formal asymptotic analysis of their solution when $d$ is large is also possible, as shown next. This analysis characterizes the asymptotic convergence rate of the eigenvalue to the target, which can be used to obtain an asymptotic convergence rate of the loss that is more precise than (19): Specifically, it shows that $E(A(t))$ eventually decays as $1/t^2$ when $m^* < d$ and exponentially fast in $t$ when $m^* \geq d$.

## 5.1 Formal asymptotic analysis of (20)

**Case $m^* \ll d$.** Then $d - m^*$ eigenvalues of $A^*$ are zero, and without loss of generality we can order $\{\lambda_i\}_{i \leq d}$ so that the zero eigenvalues of $A^*$ are last. Denoting $\epsilon(t) = \frac{1}{d-m^*}\sum_{i=m^*+1}^d \lambda_i(t)$, for $m^* < d$ (20) then reads

$$\dot{\lambda}_i = 2\Big(\sum_{j=1}^{m^*} (\lambda_j^* - \lambda_j) - (d - m^*)\epsilon\Big)\lambda_i + 4(\lambda_i^* - \lambda_i)\lambda_i, \qquad i = 1, \ldots, m^* \tag{22}$$

$$\dot{\epsilon} = 2\Big(\sum_{j=1}^{m^*} (\lambda_j^* - \lambda_j) - (d - m^*)\epsilon\Big)\epsilon - \frac{4}{d - m^*}\sum_{j=m^*+1}^d \lambda_j^2, \qquad \epsilon(0) = 1. \tag{23}$$

We will call the first $m^*$ eigenvalues $\lambda_i$ *informative eigenvalues* and the remaining $d - m^*$ (captured by $\epsilon(t)$) *non-informative eigenvalues*. We make two observations. Since $\lambda_i(0) = \epsilon(0) = 1$, initially the leading order term in the equation for the uninformative eigenvalues $\epsilon(t)$ is

$$\dot{\epsilon} \approx -2d\epsilon^2 \qquad \Rightarrow \qquad \epsilon(t) \approx \frac{1}{1 + 2dt} \qquad t \ll 1/d \tag{24}$$

Substituting this solution into (22) we deduce

$$\frac{d}{dt}\log\lambda_i \approx -2d\epsilon(t) \approx -\frac{2d}{1+2dt} \qquad \Rightarrow \qquad \lambda_i(t) \approx \frac{1}{1+2dt} \tag{25}$$

(24) and (25) imply an initial decreases in time of both non-informative and the informative eigenvalues. However, when $2d/(1+2dt)$ becomes of order one or smaller, the other terms in equation (22) take over and allow the informative eigenvalues to bounce back up. This happens at at time $t_0 = O(1)$ in $d$. Afterwards the informative eigenvalues emerge from the non-informative ones with an exponential growth, $\lambda_j(t) \sim \frac{1}{2d}e^{(2m^*+4)t}$. As a result, these informative eigenvalues eventually match the eigenvalues of the teacher at a typical time of order $t_J \sim \frac{1}{2m^*+4}\log(2d)$. This analysis also implies a quadratic decay in time of the loss at long times

$$E(A(t)) \sim 1/(16t^2) \qquad \text{as } t \to \infty. \tag{26}$$

In Sec. B we give additional details comparing the asymptotic analysis to the real dynamics when $m^* \leq d$ but not necessarily much smaller. This analysis can e.g. be done quite explicitly when the unit in the teacher are orthonormal. It indicates that $\epsilon(t) \approx 1/[1+2(2+d-m^*)t]$ at all times, and as a result shows that

$$E[A(t)] \approx \frac{1}{4}\left(\frac{d-m^*}{1+2(2+d-m^*)t}\right)^2 \tag{27}$$

at all times.

**Case with $m^* \geq d \gg 1$.** Then (20) can be written as

$$\frac{d}{dt}\log\lambda_i = 4\lambda_i^* + 2\sum_{j=1}^{d}\lambda_j^* - 4\lambda_i - 2\sum_{j=1}^{d}\lambda_j, \qquad i = 1, \ldots, d \tag{28}$$

which gives an exponential convergence to the target $A^*$, and consequently an exponential convergence in the population loss. For example, let us specialize to the case of a teacher with orthonormal hidden vectors, $\lambda_j^* = 1$ for $j = 1, \ldots, \min(m^*, d)$. The eigenvalues will converge to their target value as $|\lambda_j(t) - \lambda_j^*| \sim \frac{1}{2d}e^{-(2d+4)t}$. Consequently the loss (21) will converge to zero exponentially in this case

$$E[A(t)] \sim \frac{1}{2d}e^{-2(2d+4)t} \qquad \text{as } t \to \infty. \tag{29}$$

The results above are confirmed in the numerics. The cases when $m^* < d$ and $m^* \geq d$ are shown by the first two and last two panels in Fig. 2, respectively. When $m^* < d$ the decay of the empirical loss is quadratic, consistent with (26). In contrast, when $m^* \geq d$, the absence of non-informative eigenvalues removes the dominating terms in the loss (21). Therefore the loss is dominated by the informative eigenvalues and decays exponentially, consistent with (29). This can be clearly observed in Fig. 2, where the four panels show the population loss using teachers with $m^* = 2, 4, 8, 16$ and $d = 8$. The black dotted shows the quadratic asymptotic decay predicted in (26). The last two panels of the sequence show the exponential decay as predicted predicted in (29)

Fig. 3 shows the training and the population loss observed in the simulation using input dimension $d = 8$ and a teacher with $m^* = 1$ hidden unit. In this case our analysis suggests that the typical realization will converge to zero generalization error if $\alpha > \alpha_c = 1.875$. This can be observed on the right panel of the Fig. 3. We used a dashed line to represent the gradient in the population loss (8) and used a dotted line to represent the approximated result (27), observing the two being almost indistinguishable in the figure.

## 6 Probing the Loss Landscape with the String Method

Finally, let us show that we can use the string method [26–28] to probe the geometry of the training loss landscape and confirm numerically Theorem 3.1. The string method consists in connecting the student and the teacher with a curve (or string) in matrix space, and evolve this curve by GD while controlling its parametrization. In practice, this can be done efficiently by discretizing the string into equidistant images or points (with the Frobenius norm as metric), and iterating upon (i) evolving these images by the descent dynamics, and (ii) reparamterizing the string to make the images equidistant

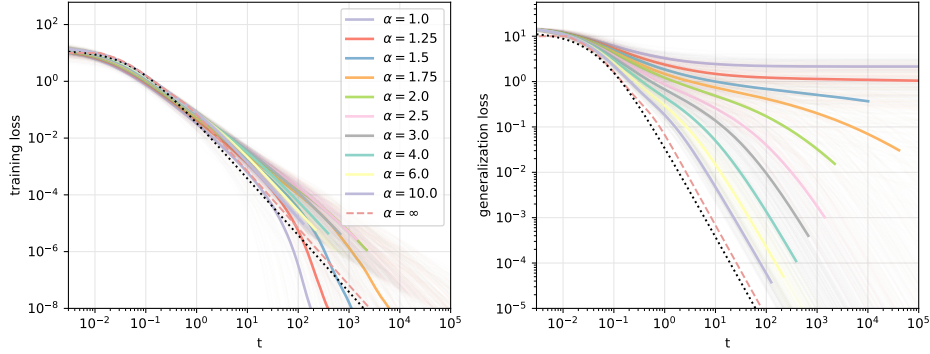

Figure 3: Training loss (left figure) and population loss (right figure) for $d = 8$ and $m^* = 1$. The plots show the average in log-scale of 100 simulation for each value of $\alpha$ and the individual realizations are shown in transparency. The results are compared with the descent in the population loss Eq. (8) (dashed pink line) and its approximation Eq. (27) (black dotted line).

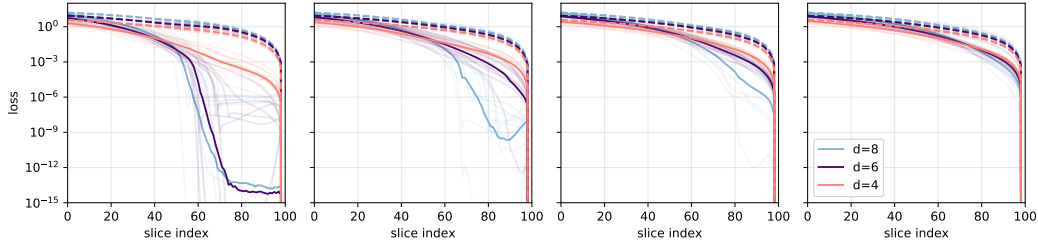

Figure 4: Results from the application of the string method. Training loss (solid line) and population loss (dashed line) evaluated across a string discretized with 100 images. Moving from left to right panels, the number of samples in the dataset increases, respectively $n = 8, 12, 16, 20$, while the teacher always has $m^* = 1$ hidden units. The critical size to obtain a smooth landscape in average is $n = 2d - 1$, which is confirmed by the string reaching zero empirical loss at a finite value of the population loss, or not. Each string is mediated in log-scale over 10 realizations.

again. At convergence the string will identify a minimum energy path between $A(0)$ and $A^*$ which will possibly have a flat portion at zero empirical loss if this loss can be minimized by GD before reaching $A^*$. That is, along the string, the student $A$ reaches the first minimum $A_\infty$ by GD, and, if $A_\infty \neq A^*$, then move along the set of minimizers of the empirical loss until it reaches $A^*$. The advantage of the method is that by replacing the physical time along the trajectory by the arclenght along it, it permits to go to infinite times (when $A = A^\infty$) and beyond (when $A^\infty \neq A^*$), thereby probing features of the loss landscape not accessible by standard GD. (Of course it requires one to know the target $A^*$ in advance, i.e. the string method cannot be used instead of GD to identify this target in situations where it is unknown.)

In Fig. 4 we compare the strings obtained for input dimension 4 (red), 6 (purple), end 8 (blue). The strings are parametrized by 100 points represented on the horizontal axes. Moving from the leftmost to the rightmost panels in Fig. 4 the number of samples in the dataset increases, namely $n = 8, 12, 16, 20$. Gradually all the $d$ represented will reach the critical size $2d - 1$ and will have a landscape with a single minimum, the informative one. Observe that for relatively small sample sizes, there is low correspondence between the topology of the training loss landscape and the population loss one. As the size increases we notice that correlation increases until the two are just slightly apart.

## Acknowledgments and Disclosure of Funding

We thank Joan Bruna and Ilias Zadik for precious discussions. SSM acknowledges the Courant Institute for the hospitality during his visit. We acknowledge funding from the ERC under the European Union's Horizon 2020 Research and Innovation Programme Grant Agreement 714608-SMiLe. We also acknowledge IPAM support from the National Science Foundation (Grant No. DMS-1440415).

## Broader Impact

Our work is theoretical in nature, and as such the potential societal consequence are difficult to foresee. We anticipate that deeper theoretical understanding of the functioning of machine learning systems will lead to their improvement in the long term.

## Footnotes

† Université Paris-Saclay, CNRS, CEA, Institut de physique théorique, 91191, Gif-sur-Yvette,France.

‡ Courant Institute, New York University, 251 Mercer Street, New York, New York 10012, USA.

§ SPOC laboratory, EPFL, Switzerland.

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
