[Supplementary Material]

# Appendix of
# Optimization and Generalization of Shallow Neural Networks with Quadratic Activation Functions

**Stefano Sarao Mannelli**[†‡*], **Eric Vanden-Eijnden**[‡], and **Lenka Zdeborová**[†]

## A   Proofs and Technical Lemmas

### A.1   Proof of Lemma 2.1

Inserting $A(t)$ as defined in (4) into (3) we arrive at

$$
\dot{A}(t) = \frac{1}{m} \sum_{i=1}^{m} \mathbb{E}_{\nu_n} \left[ \operatorname{tr} \left( X(A^* - A(t)) \right) \left( X \boldsymbol{w}_i(t) \boldsymbol{w}_i^T(t) + \boldsymbol{w}_i(t) \boldsymbol{w}_i^T(t) X \right) \right] \tag{A.1}
$$

$$
= \mathbb{E}_{\nu_n} \left[ \operatorname{tr} \left( X(A^* - A(t)) \right) \left( X A(t) + A(t) X \right) \right]
$$

where we used $X^T = X$. This proves that $\dot{A}(t)$ is equal the the rightmost equation in (5). To prove the first equality, simply note that, from (6),

$$
\nabla E_n(A) = -\mathbb{E}_{\nu_n} \left[ \operatorname{tr} \left( X(A^* - A(t)) \right) X \right] \tag{A.2}
$$

which shows that (A.1) can be written as $\dot{A} = -A \nabla E_n - \nabla E_n A$.  □

### A.2   Proof of Lemma 2.2

Equations (8) and (9) can be derived from (5) and (6) by taking their expectation over $\nu$, owing to the fact that the data is Gaussian and using Wick's theorem which asserts that

$$
\mathbb{E}_{\nu}[X_{i,j} X_{k,l}] = \delta_{i,j} \delta_{k,l} + \delta_{i,k} \delta_{j,l} + \delta_{i,l} \delta_{j,k} \tag{A.3}
$$

This gives the result since $A^*$ and $A(t)$ are symmetric matrices. Note that this derivation can be generalized to non-Gaussian data, see Ref. [1] for details.  □

### A.3   Proximal scheme

We note that (5) (and similarly (8) if we use the population loss in (9) instead of the empirical loss in (6)) can be viewed as the time continuous limit of a simple proximal scheme involving the Cholesky decomposition of $A$ and the standard Forbenius norm as Bregman distance. We state this result as:

**Proposition A.1.** *Given $B_0 \in \mathbb{R}^{d \times d}$ define the sequence of matrices $\{B_p\}_{p \in \mathbb{N}}$ via*

$$
B_p \in \arg\min_{B} \left( \frac{2}{\tau} \operatorname{tr} \left( (B - B_{p-1}) (B - B_{p-1})^T \right) + E_n(BB^T) \right) \tag{A.4}
$$

*where $\tau > 0$ is a parameter. Then*

$$
B_p B_p^T \to A(t) \qquad \text{as } \tau \to 0, \ p \to \infty \ \text{with } p\tau \to t \tag{A.5}
$$

*where $A(t)$ solves (5) for the initial condition $A(0) = B_0 B_0^T$.*

† Université Paris-Saclay, CNRS, CEA, Institut de physique théorique, 91191, Gif-sur-Yvette,France.
‡ Courant Institute, New York University, 251 Mercer Street, New York, New York 10012, USA.
∗ Work done while visiting at Courant Institute.

*Proof.* Look for a solution to the minimization problem in (A.4) of the form

$$B = B_{p-1} + \tau \tilde{B}$$

To leading order in $\tau$, the objective function in (A.4) becomes

$$\frac{2}{\tau} \operatorname{tr}\left((B - B_{p-1})(B - B_{p-1})^T\right) + E_n(BB^T)$$

$$= 2\tau \operatorname{tr}(\tilde{B}\tilde{B}^T) + \tau \operatorname{tr}\left(\left(B_{p-1}\tilde{B}^T + \tilde{B}B_{p-1}^T\right)\nabla E_n(B_{p-1}B_{p-1}^T)\right) + O(\tau^2)$$

$$= \tau \operatorname{tr}\left(\tilde{B}\left(\tilde{B}^T + B_{p-1}^T \nabla E_n(B_{p-1}B_{p-1}^T)\right)\right)$$

$$+ \tau \operatorname{tr}\left(\left(\tilde{B} + \nabla E_n(B_{p-1}B_{p-1}^T)B_{p-1}\right)\tilde{B}^T\right) + O(\tau^2)$$

which we can set to zero by choosing $\tilde{B} = \tilde{B}_p$ with

$$\tilde{B}_p = -\nabla E_n(B_{p-1}B_{p-1}^T)B_{p-1} + O(\tau)$$

In terms of the minimizer $B_p$ of the orginal problem this equation can be written as

$$\tau^{-1}(B_p - B_{p-1}) = -\nabla E_n(B_{p-1}B_{p-1}^T)B_{p-1} + O(\tau)$$

Letting $\tau \to 0$ and $p \to \infty$ with $p\tau \to t$, we deduce that $B_p \to B(t)$ solution to

$$\dot{B}(t) = -\nabla E_n(B(t)B^T(t))B(t) \tag{A.6}$$

Setting $A(t) = B(t)B^T(t)$ we have

$$\dot{A}(t) = \dot{B}(t)B^T(t) + B(t)\dot{B}^T(t)$$

$$= -\nabla E_n(B(t)B^T(t))B(t)B^T(t) - B(t)B^T(t)\nabla E_n(B(t)B^T(t))$$

$$= -\nabla E_n(A(t))A(t) - A(t)\nabla E_n(A(t))$$

which is (5). $\qquad \square$

## A.4  Proof of Theorem 3.1

Let $A_{n,d}$ be a symmetric, positive semidefinite minimizer of the empirical loss and consider $A_{n,d} - A^*$. Since this matrix is symmetric, there exists an orthonormal basis in $\mathbb{R}^d$ made of its eigenvectors, $\{v_i\}_{i=1}^d$. Since $A_{n,d}$ is positive semidefinite by assumption and $A^* = w^*(w^*)^T$ is rank one, $d-1$ eigenvalues of $A_{n,d} - A^*$ are nonnegative, and only one can be positive, negative, or zero. Let us order the eigenvectors $v_i$ such that their associate eigenvalues are $\lambda_i \geq 0$ for $i = 1, \ldots, d$ and $\lambda_d \in \mathbb{R}$. Given the data vector $\{x_k\}_{k=1}^n$, to be a minimizer of the empirical loss $A_{n,d}$ must satisfy

$$\forall k = 1, \ldots, n \quad : \quad 0 = \operatorname{tr}[X_k(A_{n,d} - A^*)] = \langle x_k, (A_{n,d} - A^*)x_k \rangle = \sum_{i=1}^d \lambda_i |x_k \cdot v_i|^2 \tag{A.7}$$

Let us analyze when (A.7) admits solutions that are not $A^*$. To this end, assume first that $\lambda_d \geq 0$. Then, as soon as $n \geq d$, for each $i \in \{1, \ldots, d\}$ with probability one there is at least one $k \in \{1, \ldots, n\}$ such that $x_k \cdot v_i \neq 0$. As a result, if $\lambda_d \geq 0$, as as soon as $n \geq d$, the only solution to (A.7) is $\lambda_i = 0$ for all $i = 1, \ldots, d$, i.e. $A_{n,d} = A^*$ a.s.

The worst scenario case is actually when $\lambda_d < 0$. In that case (A.7) can be written

$$\forall k = 1, \ldots, n \quad : \quad \sum_{i=1}^{d-1} \lambda_i |x_k \cdot v_i|^2 = |\lambda_d||x_k \cdot v_d|^2 \tag{A.8}$$

This equation means that if we let $\hat{x}_k = x_k \operatorname{sign}(x_k \cdot v_d)$ (i.e. $\hat{x}_k \parallel x_k$ but lie in the same hemisphere as $v_d$), then the vectors $\hat{x}_k$ must all lie on the surface of an elliptical cone $C$ centered around $v_d$, with the principal axes of the ellipsoids aligned with $v_i$, $i = 1, \ldots, d-1$; the intersection of the cone with the hyperplane $x \cdot v_d = 1$ is the $d-1$ ellipsoid whose boundary satisfies the equation

$$\sum_{i=1}^{d-1} \lambda_i |x \cdot v_i|^2 = |\lambda_d| \tag{A.9}$$

In $\mathbb{R}^d$, it takes $\frac{1}{2}d(d+1)$ vectors $\hat{\boldsymbol{x}}_k$ to uniquely define such a elliptical cone. This means that, in the worst case scenario, we recover the threshold $n = \frac{1}{2}d(d+1)$. This worst case scenario is however unlikely. To see why, assume that $n \geq d$, and consider the convex polyhedral cone spanned by $\{\hat{\boldsymbol{x}}_k\}_{k=1}^n$, i.e. the region

$$C_{n,d} = \{\boldsymbol{x} : \boldsymbol{x} = \sum_{k=1}^n \alpha_k \hat{\boldsymbol{x}}_k, \alpha_k \geq 0, k = 1, \ldots, n\} \subset \mathbb{R}^d \tag{A.10}$$

In order that (A.9) have a nontrivial solution, the extremal rays of $C_{n,d}$ (i.e. its edges of dimension 1) must coincide with the set $\{\hat{\boldsymbol{x}}_k\}_{k=1}^n$, that is, all rays $\alpha_k \hat{\boldsymbol{x}}_k$, $\alpha_k \geq 0$ for $k = 1, \ldots, n$ must lie on the boundary of $C_{n,d}$ and none can be in the interior of $C_{n,d}$; indeed these extremal rays must also be on the boundary of elliptical cone $C$. However, Theorem 3' in [2] asserts that, if the vectors in the set $\{\hat{\boldsymbol{x}}_k\}_{k=1}^n$ are in general position (i.e. if the vectors in any subset of size no more than $d$ are linearly independent, which happens with probability one if $\boldsymbol{x}_k$ are i.i.d. Gaussian), the number $N_{n,d}$ of extremal rays of $C_{n,d}$ satisfies

$$\mathbb{E}_\nu N_{n,d} = 2n \frac{C(n-1, d-1)}{C(n,d)}, \qquad C(n,d) = 2 \sum_{k=0}^{d-1} \binom{n-1}{k} \tag{A.11}$$

This implies that

$$\lim_{d \to \infty} d^{-1} \mathbb{E}_\nu N_{\lfloor \alpha d \rfloor, d} = \begin{cases} \alpha & \text{if } \alpha \in [1, 2] \\ 2 & \text{if } \alpha \in (2, \infty) \end{cases} \tag{A.12}$$

Since $N_{n,d} \leq n$ by definition, we have $d^{-1} N_{\lfloor \alpha d \rfloor, d} \leq \alpha$, which from (A.12) implies that $\lim_{d \to \infty} d^{-1} N_{\lfloor \alpha d \rfloor, d} = \alpha$ a.s. if $\alpha \in [1, 2]$. In turns this implies that the probability that all the vectors in $\{\hat{\boldsymbol{x}}_k\}_{k=1}^n$ be extremal ray of the cone $C_{n,d}$ tends to 1 as $d, n \to \infty$ with $n = \lfloor \alpha d \rfloor$ and $\alpha \in [0, 2]$. This also means that the probability that (A.9) has solution with $\lambda_d < 0$ also tends to 1 in this limit, i.e. (11) holds. Conversely, since $\lim_{d \to \infty} d^{-1} N_{\lfloor \alpha d \rfloor, d} = 2 < \alpha$ for $\alpha > 2$, the probability that $N_{n,d} \neq n$ remains positive as $d, n \to \infty$ with $n = \lfloor \alpha d \rfloor$ and $\alpha \in (2, \infty)$. This means that the probability that (A.9) has no solution with $\lambda_d < 0$ is positive in this limit, i.e. (12) holds. $\qquad\square$

## A.5 Heuristic argument for arbitrary $m^*$ and $d$

Minimizers of the empirical loss satisfy:

$$\forall k = 1, \ldots, n \quad : \quad \text{tr}[X_k(A - A^*)] = \langle \boldsymbol{x}_k, (A - A^*)\boldsymbol{x}_k \rangle = 0 \tag{A.13}$$

Clearly $A = A^*$ is always a solution to this set of equation. The question is: how large should $n$ be in order that $A = A^*$ be the only solution to that equation? If $A$ was an arbitrary symmetric matrix, we already know the answer: with probability one, we need $n \geq \frac{1}{2}d(d+1)$. What makes the problem more complicated is that $A$ is required positive semidefinite. If we assume that $A^*$ has rank $m^* < d$, this implies that $C = A - A^*$ must be a symmetric matrix with $d - m^*$ nonnegative eigenvalues and $m^*$ eigenvalues whose sign is unconstrained, and we need to understand what this requirement imposes on the solution to (A.13).

In the trivial case when $m^* = 0$ (i.e. $A^* = 0$), if we decompose $A = U\Lambda U^T$, where $U$ contains its eigenvectors and $\Lambda$ is a diagonal matrix with its eigenvectors $\lambda_i \geq 0$, $i = 1, \ldots, d$, (A.13) can be written as

$$\forall k = 1, \ldots, n \quad : \quad \sum_{i=1}^d \lambda_i (\boldsymbol{v}_i \cdot \boldsymbol{x}_k)^2 = 0 \tag{A.14}$$

where $\boldsymbol{v}_i$, $i = 1, \ldots, d$ are linearly independent eigenvectors of $A$. In this case, since $\lambda_i \geq 0$, with probability one we only need $n = d$ data vectors to guarantee that the only solution to this equation is $\lambda_i = 0$ for all $i = 1, \ldots, d$, i.e. $A = 0$. Another way to think about this is to realize that the nonnegativity constraint on $A$ has removed $\frac{1}{2}d(d-1)$ degrees of freedom from the original $\frac{1}{2}d(d+1)$ in $A$.

If $m^* > 0$, the situation is more complicated, but we can consider the projection of $A$ in the subspace not spanned by $A^*$, i.e. the $(d - m^*) \times (d - m^*)$ matrix $A^\perp$ defined as

$$A^\perp = (V^*)^T A V^* \tag{A.15}$$

where $V^*$ is the $d \times (d - m^*)$ matrix whose columns are linearly independent eigenvectors of $A$ with zero eigenvalue. All the eigenvalues of $A^\perp$ are nonnegative, and this imposes $\frac{1}{2}(d - m^*)(d - m^* - 1)$

constraints in the subspace where $A^\perp$ lives. If we simply subtract this number to $\frac{1}{2}d(d+1)$ we obtain

$$n_c = \tfrac{1}{2}d(d+1) - \tfrac{1}{2}(d-m^*)(d-m^*-1) = d(m^*+1) - \tfrac{1}{2}m^*(m^*+1) \tag{A.16}$$

which is precisely (13).

This argument is nonrigorous because we cannot *a priori* treat separately (A.13) in the subspace spanned by $A^*$ and its orthogonal complement. Yet, our numerical results suggest that this assumption is valid, at least as $d \to \infty$.

## A.6  Proof of Theorem 4.1

Since (3) is a standard gradient flow, $\dot{\boldsymbol{w}}_i(t) = -m\partial_{\boldsymbol{w}_i}L_n$ and the loss is a quartic polynomial in the weights, we can invoke the Stable Manifold Theorem [3] to conclude that the stable manifolds of local minimizers of $L_n$ have codimension 0, whereas the stable manifolds of all other critical points have codimension strictly larger than 0. As a result, the weights must converge towards a local minimizer of the loss with probability one with respect to random initial data drawn for any probability distribution that is absolutely continuous with respect to the Lebesgue measure on $\mathbb{R}^{md}$: this is the case under our assumption on $\{\boldsymbol{w}_i(0)\}_{i=1}^m$. Denoting $\boldsymbol{w}_i^\infty = \lim_{t\to\infty}\boldsymbol{w}_i(t)$, since $\{\boldsymbol{w}_i^\infty\}_{i=1}^m$ is a local minimizer of $L_n$, there exits $\delta > 0$ such that

$$\frac{1}{m}\sum_{i=1}^m |\boldsymbol{w}_i - \boldsymbol{w}_i^\infty|^2 \leq \delta \qquad \Rightarrow \qquad L_n(\boldsymbol{w}_i) \geq L_n(\boldsymbol{w}_i^\infty) \tag{A.17}$$

Since the GD flow in (3) for the weights implies (5) as evolution equation for $A(t) = m^{-1}\sum_{i=1}^m \boldsymbol{w}_i(t)\boldsymbol{w}_i^T(t)$ by Lemma 2.1, it follows that $\lim_{t\to\infty} A(t) = A_\infty = m^{-1}\sum_{i=1}^m \boldsymbol{w}_i^\infty \boldsymbol{w}_i^\infty$. As soon as $m \geq d$, any symmetric positive semidefinite $A$ can be constructed via a set of weights $\{\boldsymbol{w}_i\}_{i=1}^m$, i.e.

$$\forall A = A^T \text{ PSD} \quad \exists\{\boldsymbol{w}_i\}_{i=1}^m \quad : \quad A = \frac{1}{m}\sum_{i=1}^m \boldsymbol{w}_i\boldsymbol{w}_i^T \quad \& \quad L_n(\boldsymbol{w}_1,\ldots,\boldsymbol{w}_m) = 2E_n(A) \tag{A.18}$$

This implies that $A_\infty$ must be a local minimizer of the empirical loss $E_n(A)$, otherwise for any $\epsilon > 0$ there would be a $A$ such that

$$\text{tr}\left[(A-A_\infty)^2\right] \leq \epsilon \quad \& \quad E_n(A) < E_n(A_\infty) \tag{A.19}$$

Choosing $\{\boldsymbol{w}_i\}_{i=1}^m$ such that $A = \frac{1}{m}\sum_{i=1}^m \boldsymbol{w}_i\boldsymbol{w}_i^T$ would contradict (A.17). This also implies (16) since all the minimizers of the empirical loss are global minimizers by convexity, and $E_n(A_\infty) = E_n(A^*) = 0$. $\qquad\square$

## A.7  Proof of Theorem 5.1

We begin with:

*Proof of* (17) *in Theorem 5.1.* We can follow the same steps as in the proof of Theorem 4.1, using (7) instead of (3), and noticing that this equation is also a standard gradient flow, $\dot{\boldsymbol{w}}_i(t) = -m\partial_{\boldsymbol{w}_i}L$ on the quartic loss

$$L(\boldsymbol{w}_1,\ldots,\boldsymbol{w}_m) = 2E(A) \qquad \text{with } A = \frac{1}{m}\sum_{i=1}^m \boldsymbol{w}_i\boldsymbol{w}_i^T \tag{A.20}$$

and $E(A)$ given in 9. The only difference is that the minimizer of $E(A)$ is now unique and given by $A^*$, which guarantees (17). $\qquad\square$

This leaves us with establishing the convergence rates in (18) and (19). Their proof replies on three Lemmas that we state first.

**Lemma A.2.** *Let $A(t)$ be the solution to the GD flow* (8) *and assume that $A(0)$ has full rank. Then we have*

$$A^{-1}(t) = \mathbb{E}[\boldsymbol{z}(t)\boldsymbol{z}^T(t)] \tag{A.21}$$

*where $\boldsymbol{z}(t) \in \mathbb{R}^d$ solves the stochastic differential equation (SDE)*

$$d\boldsymbol{z} = (\operatorname{tr}(A - A^*))\boldsymbol{z}dt - 2A^*\boldsymbol{z}dt + 2d\boldsymbol{W}(t) \tag{A.22}$$

*Here $\boldsymbol{W}(t)$ is a standard $d$-dimensional Wiener process and we impose that the initial condition $\boldsymbol{z}(0)$ be Gaussian, independent of $\boldsymbol{W}$, with mean zero and covariance $\mathbb{E}[\boldsymbol{z}(0)\boldsymbol{z}^T(0)] = A^{-1}(0)$.*

**Lemma A.3.** *Under the conditions of Lemma A.2, we have the following identity for all $t \geq \tau \geq 0$:*

$$A^{-1}(t) = e^{-2A^*(t-\tau)}A^{-1}(\tau)e^{-2A^*(t-\tau)}\exp\left(2\int_\tau^t \operatorname{tr}(A(s) - A^*)ds\right)$$
$$+ 4\int_\tau^t e^{-4A^*(t-s)}\exp\left(2\int_s^t \operatorname{tr}(A(u) - A^*)du\right)ds \tag{A.23}$$

**Lemma A.4.** *Under the conditions of Lemma A.2, we have*

$$\lim_{t\to\infty} \operatorname{tr}(A^*A^{-1}(t)A^*) = \operatorname{tr} A^* \tag{A.24}$$

Note that, if $m^* \geq d$ and $A^*$ is invertible, since $\lim_{t\to\infty} A(t) = A^*$, we have $\lim_{t\to\infty} A^{-1}(t) = (A^*)^{-1}$ and (A.24) trivially holds. This equation also holds when $m^* < d$, i.e. when $A^*$ is rank deficient and not invertible, if we assume that $A(0) = \operatorname{Id}$ so that $A(t)$ remains diagonal at all times by Theorem 5.2 since, using the notations of this theorem, we then have

$$\lim_{t\to\infty} \operatorname{tr}(A^*A^{-1}(t)A^*) = \lim_{t\to\infty} \sum_{i=1}^{m^*} \frac{(\lambda_i^*)^2}{\lambda_i(t)} = \sum_{i=1}^{m^*} \lambda_i^* = \operatorname{tr} A^* \tag{A.25}$$

because $\lim_{t\to\infty} \lambda_i(t) = \lambda_i^* > 0$ if $i \leq m^*$. However, the dangerous case is when $m^* < d$ and $A(t)$ is not diagonal: in that case (A.24) is nontrivial.

*Proof of Lemma A.2.* Since $A(0)$ has full rank, $A^{-1}(0)$ exists, and since $A(t)$ solves (8), $A^{-1}(t)$ satisfies

$$\frac{d}{dt}A^{-1}(t) = 2\left[(\operatorname{tr}(A - A^*))A^{-1} + A^{-1}(A - A^*) + (A - A^*)A^{-1}\right]$$
$$= 2(\operatorname{tr}(A - A^*))A^{-1} - 2A^{-1}A^* - 2A^*A^{-1} + 4\operatorname{Id} \tag{A.26}$$

A direct calculation with (A.22) using Itô formula shows that $\mathbb{E}[\boldsymbol{z}(t)\boldsymbol{z}^T(t)]$ solves (A.26) for the same initial condition, i.e. (A.21) holds. $\qquad\square$

*Proof of Lemma A.3.* Equation (A.21) implies that

$$\operatorname{tr}(A^*A^{-1}(t)A^*) = \mathbb{E}|A^*\boldsymbol{z}(t)|^2 \tag{A.27}$$

Since the solution to (A.22) can be expressed as

$$\boldsymbol{z}(t) = \exp\left(-2A^*(t - \tau) + \int_\tau^t \operatorname{tr}(A(s) - A^*)ds\right)\boldsymbol{z}(\tau)$$
$$+ 2\int_\tau^t \exp\left(-2A^*(t - s) + \int_s^t \operatorname{tr}(A(u) - A^*)du\right)d\boldsymbol{W}(s), \tag{A.28}$$

a direct calculation using this formula in (A.27) together with $\mathbb{E}[z(\tau)z^T(\tau)] = A^{-1}(\tau)$ and Itô isometry establishes (A.23). $\qquad\square$

*Proof of Lemma A.4.* We only need to consider the nontrivial case when $A^*$ is rank deficient, i.e. $m^* < d$. To begin, notice that (A.23) implies the following identity for all $t \geq \tau \geq 0$:

$$\operatorname{tr}(A^*A^{-1}(t)A^*) = \operatorname{tr}\left(A^*e^{-2A^*(t-\tau)}A^{-1}(\tau)e^{-2A^*(t-\tau)}A^*\right)\exp\left(2\int_\tau^t \operatorname{tr}(A(s) - A^*)ds\right)$$
$$+ 4\int_\tau^t \operatorname{tr}\left(A^*e^{-4A^*(t-s)}A^*\right)\exp\left(2\int_s^t \operatorname{tr}(A(u) - A^*)du\right)ds \tag{A.29}$$

Since $A^*$ is symmetric and positive semidefinite, its eigenvalues are nonnegative and there exists an orthonormal basis made of its eigenvectors. Denote this basis by $\{\boldsymbol{v}_i^*\}_{i=1}^d$ and let us order it in way

that the corresponding eigenvalues are $\lambda_i^* > 0$ for $i = 1, \ldots, m^*$, and $\lambda_i^* = 0$ for $i = m^* + 1, \ldots, d$. Then (A.29) can be written as

$$\mathrm{tr}(A^* A^{-1}(t) A^*) = \sum_{i=1}^{m^*} (\lambda_i^*)^2 (\boldsymbol{v}_i^*)^T A^{-1}(\tau) \boldsymbol{v}_i^* \exp\left(-4\lambda_i^*(t-\tau) + 2\int_\tau^t \mathrm{tr}(A(s) - A^*)ds\right)$$

$$+ 4 \sum_{i=1}^{m^*} (\lambda_i^*)^2 \int_\tau^t \exp\left(-4\lambda_i^*(t-s) + 2\int_s^t \mathrm{tr}(A(u) - A^*)du\right) ds$$

(A.30)

Since $|\mathrm{tr}(A(t) - A^*)|$ is bounded for all $t \geq 0$, evaluating this expression at $\tau = 0$ shows that $\mathrm{tr}(A^* A^{-1}(t) A^*)$ is also bounded i.e. we only need to consider what happens as $t \to \infty$. We have

$$\forall t \geq \tau \quad : \quad \left| \frac{\int_\tau^t \mathrm{tr}(A(u) - A^*)du}{2(t-\tau)} \right| \leq C(\tau) := \frac{1}{2} \max_{u \in [\tau, \infty)} |\mathrm{tr}(A(u) - A^*)| < \infty \qquad \text{(A.31)}$$

with $C(\tau)$ decaying to zero as $\tau \to \infty$ since $\lim_{t \to \infty} A(t) = A^*$ by (17). This implies that the first term at the right hand side of (A.30) can be bounded as

$$\sum_{i=1}^{m^*} (\lambda_i^*)^2 (\boldsymbol{v}_i^*)^T A^{-1}(\tau) \boldsymbol{v}_i^* \exp\left(-4\lambda_i^*(t-\tau) + 2\int_\tau^t \mathrm{tr}(A(s) - A^*)ds\right)$$

(A.32)

$$\leq \sum_{i=1}^{m^*} (\lambda_i^*)^2 (\boldsymbol{v}_i^*)^T A^{-1}(\tau) \boldsymbol{v}_i^* \exp\left(-4\lambda_i^*(t-\tau)[1 - C(\tau)/\lambda_i^*]\right).$$

Since $\lim_{\tau \to \infty} C(\tau) = 0$, there exists $\tau_c \geq 0$ such that $C(\tau) < \min_{i=1, \ldots, m^*} \lambda_i^*$ for all $\tau \geq \tau_c$, and hence $1 - C(\tau)/\lambda_i^* > 0$ for all $\tau \geq \tau_c$ and for all $i = 1, \ldots, m^*$. Therefore we can let $t \to \infty$ at any fixed $\tau \geq \tau_c$ in (A.32) to conclude that the limit of the first term at the right hand side of (A.30) is zero, i.e.

$$\lim_{t \to \infty} \sum_{i=1}^{m^*} (\lambda_i^*)^2 (\boldsymbol{v}_i^*)^T A^{-1}(\tau) \boldsymbol{v}_i^* \exp\left(-4\lambda_i^*(t-\tau) + 2\int_\tau^t \mathrm{tr}(A(s) - A^*)ds\right) = 0 \qquad (\tau \geq \tau_c).$$

(A.33)

Similarly, to deal with the second term at the right hand side of (A.30), we can use

$$\forall t \geq s \geq \tau \quad : \quad \left| \frac{\int_s^t \mathrm{tr}(A(u) - A^*)du}{2(t-s)} \right| \leq C(\tau) \qquad \text{(A.34)}$$

with the same $C(\tau)$ as in (A.31). As a result, by taking again $\tau \geq \tau_c$, we have

$$\lim_{t \to \infty} 4 \sum_{i=1}^{m^*} (\lambda_i^*)^2 \int_\tau^t \exp\left(-4\lambda_i^*(t-s) + 2\int_s^t \mathrm{tr}(A(u) - A^*)du\right) ds$$

$$\leq 4 \sum_{i=1}^{m^*} (\lambda_i^*)^2 \lim_{t \to \infty} \int_\tau^t \exp\left(-4\lambda_i^*(t-s)[1 - C(\tau)/\lambda_i^*]\right) ds \qquad \text{(A.35)}$$

$$= \sum_{i=1}^{m^*} \lambda_i^* [1 - C(\tau)/\lambda_i^*]^{-1} \qquad (\tau \geq \tau_c)$$

Therefore we have established that

$$\lim_{t \to \infty} \mathrm{tr}(A^* A^{-1}(t) A^*) \leq \sum_{i=1}^{m^*} \lambda_i^* [1 - C(\tau)/\lambda_i^*]^{-1} \qquad (\tau \geq \tau_c) \qquad \text{(A.36)}$$

Since this equation is valid for any $\tau \geq \tau_c$ and $\lim_{\tau \to \infty} C(\tau) = 0$, we can now let $\tau \to \infty$ on the right hand side of (A.36) to deduce

$$\lim_{t \to \infty} \mathrm{tr}(A^* A^{-1}(t) A^*) \leq \sum_{i=1}^{m^*} \lambda_i^* = \mathrm{tr}\, A^* \qquad \text{(A.37)}$$

To get the matching lower bound, use $\operatorname{tr}\left((A(t)-A^*)A^{-1}(t)(A(t)-A^*)\right) \geq 0$ to deduce

$$\operatorname{tr}(A^*A^{-1}(t)A^*) \geq 2\operatorname{tr}A^* - \operatorname{tr}A(t) \tag{A.38}$$

and take the limit as $t \to \infty$ using $\lim_{t\to\infty}\operatorname{tr}A(t) = A^*$ to obtain

$$\lim_{t\to\infty}\operatorname{tr}(A^*A^{-1}(t)A^*) \geq \operatorname{tr}A^* \tag{A.39}$$

Taken together (A.37) and (A.39) imply (A.24). $\qquad\square$

We can now use these results to proceed with the rest of the proof of Theorem 5.1:

*Proof of* (18) *and* (19) *in Theorem 5.1.* The multiplicative inverse of $E(A(t))$ satisfies

$$\frac{d}{dt}E^{-1}(A(t)) = 2E^{-2}(A(t))\operatorname{tr}[\nabla E(A(t))A(t)\,\nabla E(A(t))]. \tag{A.40}$$

By convexity of $E(A)$ we have

$$E(A(t)) \leq \operatorname{tr}[(A(t)-A^*)\nabla E(A(t))] = \operatorname{tr}[(A(t)-A^*)A^{-1/2}(t)A^{1/2}(t)\nabla E(A(t))] \tag{A.41}$$

where we used the positivity of $A(t)$ as well as $E(A^*) = 0$. Therefore using Cauchy-Schwarz inequality we obtain

$$E^2(A(t)) \leq \operatorname{tr}[\nabla E(A(t))A(t)\,\nabla E(A(t))]\operatorname{tr}\left[(A^*(t)-A^*)A^{-1}(t)(A(t)-A^*)\right]. \tag{A.42}$$

Using this inequality in (A.40) we deduce

$$\frac{d}{dt}E^{-1}(A(t)) \geq 2\left[\operatorname{tr}\left((A(t)-A^*)A^{-1}(t)(A(t)-A^*)\right)\right]^{-1}$$

Integrating and reorganizing gives

$$E(A(t)) \leq \frac{E(A(0))}{1 + 2E(A(0))\int_0^t \left[\operatorname{tr}\left((A(s)-A^*)A^{-1}(s)(A(s)-A^*)\right)\right]^{-1}ds} \tag{A.43}$$

To proceed let us analyze the behavior of the integral in the denominator. Start by noticing that

$$\begin{aligned}
&\lim_{t\to\infty}\operatorname{tr}\left((A(t)-A^*)A^{-1}(t)(A(t)-A^*)\right) \\
&= \lim_{t\to\infty}\operatorname{tr}A(t) - 2\operatorname{tr}A^* + \lim_{t\to\infty}\operatorname{tr}(A^*A^{-1}(t)A^*) = 0
\end{aligned} \tag{A.44}$$

where we used $\lim_{t\to\infty}\operatorname{tr}A(t) = \operatorname{tr}A^*$ as well as (A.24) in Lemma A.3. (A.44) guarantees that $\operatorname{tr}\left[(A(t)-A^*)A^{-1}(t)(A(t)-A^*)\right]$ is bounded for all time, i.e.

$$\forall t \geq 0 \quad : \quad \left[\operatorname{tr}\left((A(t)-A^*)A^{-1}(t)(A(t)-A^*)\right)\right]^{-1} \geq C > 0 \tag{A.45}$$

with

$$C = \left[\max_{t\in[0,\infty)}\operatorname{tr}\left[(A(t)-A^*)A^{-1}(t)(A(t)-A^*)\right]\right]^{-1} \tag{A.46}$$

As a result

$$\forall t \geq 0 \quad : \quad \int_0^t \operatorname{tr}\left[(A(s)-A^*)A^{-1}(s)(A(s)-A^*)\right]^{-1}ds \geq Ct \tag{A.47}$$

which from (A.43) implies the nonasymptotic bound in (18). To establish the asymptotic bound in (19), note that (A.44) implies that

$$\int_0^t \left[\operatorname{tr}\left((A(s)-A^*)A^{-1}(s)(A(s)-A^*)\right)\right]^{-1}ds \text{ grows faster than } t \text{ as } t \to \infty \tag{A.48}$$

Using this result in (A.43) implies that $E(A(t))$ decays faster than $1/t$ as $t \to \infty$, i.e. $E(A(t)) = o(1/t)$ and (19) holds. $\qquad\square$

## A.8  Proof of Theorem 5.2

Since $A^*$ is symmetric and positive semidefinite, its eigenvalues are nonnegative and there exists an orthonormal basis made of its eigenvectors. Denote this basis by $\{v_i^*\}_{i=1}^d$ and let us order it in way that the corresponding eigenvalues are $\lambda_i^* > 0$ for $i = 1, \ldots, m^*$, and $\lambda_i^* = 0$ for $i = m^* + 1, \ldots, d$. Denote by $U^*$ the orthogonal matrix whose columns are the eigenvectors of $A^*$, so that $A^* = U^* \Lambda^* (U^*)^T$ with $\Lambda^* = \mathrm{diag}(\lambda_1^*, \ldots, \lambda_d^*)$. Let $\Lambda(t) = (U^*)^T A(t) U^*$. Since $A(0) = \mathrm{Id}$ by assumption, $\Lambda(0) = \mathrm{Id}$ and from (8) this matrix evolves according to

$$\dot{\Lambda} = 2(\mathrm{tr}(A^* - A))(U^*)^T A U^* + 2(U^*)^T (A^* - A) A U^* + 2(U^*)^T A (A^* - A) U^* \\ = 2(\mathrm{tr}(\Lambda^* - \Lambda))\Lambda + 2\Lambda(\Lambda^* - \Lambda) + 2(\Lambda^* - \Lambda)\Lambda. \tag{A.49}$$

This equation shows that $\Lambda(t)$ remains diagonal for all times, $\Lambda(t) = \mathrm{diag}(\lambda_1(t), \ldots, \lambda_d(t))$. Written componentwise (A.49) is (20). $\qquad\square$

# B  Additional Results

## B.1  Supporting numerical results to Fig. 1

Figure B.1: Population loss for $d = 4, 8, 16$ and $m^* = 4$ and several values of $\alpha = n/d$. The line shown with full color are average of the logarithm of 100 simulations (300 for $d = 4$) and the individual instances are shown in transparency.

In Fig. B.1 shows the average performance of GD with $n = \alpha d$ datapoints and a teacher with $m^*$ and Gaussian hidden units. The figure is intended to show a vertical cut in the dynamical phases Fig. 1. Moving up in $d$ at $m^*$ fixed we observe that on average the simulations converge when $\alpha > \alpha_c$ and they do not when $\alpha > \alpha_c$, i.e. there is an abrupt change of behavior when we cross the transition. Another interesting aspect of the figure is that the first panel has $m^* \geq d$ which leads to and exponential (rather than quadratic) convergence rate in the loss, consistent to our analysis. The dotted line is a reference line that represents the $1/t^2$ decay of the loss.

## B.2 Supporting numerical results to Theorem 3.1

Figure B.2: Left panel: fraction of simulations that went below $10^{-5}$ for $d = 4, 8, 16, 32$. Right panel: complement of the fraction of simulations that have a ratio between final generalization loss and training loss that is larger then $10^9 d$.

In Fig. B.2 we present a numerical verification of Theorem 3.1. According to the theorem, as $d \to \infty$ with $m^* = 1$ (so that $\alpha_c = 2$) the probability of finding the teacher should converge to zero for $\alpha < 2$ and to positive values for $\alpha > 2$. The left panel on the figure shows the fraction of 100 simulations that achieved at least $10^{-5}$ generalization loss after $2 \log_2 d \times 10^7$ iterations with learning rate $0.003$. The right panel shows the number of simulations for which the ratio between training and generalization losses is larger than $10^{-9} d^{-1}$. This second panel is meant to capture the simulations for which we expect convergence eventually, but the number of iterations was not enough to achieve it. In particular, we observed that when generalization fails, meaning that the training loss goes to zero and the generalization loss stay at a high value, the convergence rate of the training loss is exponential, contrarily to simulation where the generalization loss eventually goes to zero that have a $O(1/t^2)$ convergence rate. Using simulations with $10^8$ iterations is sufficient to detect the difference between the two cases and therefore this gives us a good criterion to distinguish between successful and unsuccessful simulations.

Figure B.3: Final value of the training and generalization loss of several simulations with input $d = 4$ and $n = 7$ samples in the dataset. From left to right the maximum number of steps in the simulation increases by a factor 10.

To provide more evidence of this reasoning, in Fig. B.3 we show training and generalization loss of 1000 simulations for $d = 4$, $n = 2d - 1$ and $m^* = 1$. We order the simulations according to the loss and show in the three panels three snapshots for different number of iterations. From left to right the number of iterations increases by a factor 10 in each panel. As can be seen, the ratio between generalization loss and training loss at the end of the training is a valid measure of success.

## B.3 Extrapolation procedure

Figure B.4: Extrapolation of the sample complexity threshold $\alpha_c = 2$ for $m^* = 1$ assuming a power-law increase of the time to converge to a $10^{-5}$ value of the loss when approaching this threshold. In the inset we show that the points lie on a line in log-log scale.

We estimate the critical value of $\alpha$ numerically by fixing a threshold in the population loss, $10^{-5}$, and simulate the problem for a large set of $\alpha$. Starting from the largest value in the set, as $\alpha$ approaches the critical value the time needed to pass the threshold increase as a power-law $\sim |\alpha - \alpha_c|^{-\theta}$ [4]. In Fig. B.4 we fit the relaxation times to cross a threshold in the population loss of $10^{-5}$ for $d = 4, 8, 16, 32$ and $m^* = 1$. The extrapolated thresholds $\alpha_c$ and their 95% confidence intervals are: for $d = 4$, $\alpha_c = 1.6$ $(1.3, 1.9)$; for $d = 8$, $\alpha_c = 1.8$ $(1.4, 2.2)$; for $d = 16$, $\alpha_c = 2.2$ $(2.0, 2.5)$; and for $d = 32$, $\alpha_c = 2.4$ $(2.0, 2.8)$. Close to the threshold $\alpha_c = 2 - 1/d$, namely 1.8, 1.9, 1.9, and 2.0, as expected. The larger the input dimension, the larger the time to pass the threshold is, and as result the smallest accessible value of $\alpha$ also increases. This causes a decrease in accuracy on the threshold value, measured by the larger confidence intervals obtained assuming a t-student distribution. The same procedure has been applied for other values of $m^*$ to obtain the points shown in Fig. 1.

## B.4 GD in the populations loss with orthogonal teacher

Figure B.5: Same as Fig. 1 in the main text but for a teacher with orthonormal hidden nodes. In that case, as soon as $m^*$ becomes equal to or larger than $d$, $A^* = $ Id, and therefore the student equal the teacher at initialization since $A(0) = $ Id.

A simple special case of (20) in Theorem 5.2 is when the teacher has orthogonal hidden weights, so that $\lambda_j^* = 1$ for $1 \leq j \leq m^*$ and $\lambda_j^* = 0$ for every $m^* < j < d$. (Note that the problem becomes

Figure B.6: Phase portrait of the the Lotka-Volterra system in (B.1)-(B.2) both in linear (left panel) and log (right panel) scales, for $d = 16$ and $m^* = 1$. The $\lambda$- and $\epsilon$-nullclines are shown in red and orange, respectively. The flow map is show in black. The actual solution starting from $(\lambda(0), \epsilon(0)) = (1, 1)$ is shown in blue.

trivial in that case when $m^* = d$ since $A(t = 0) = \text{Id} = A^*$.) In that case the first $m^*$ *informative eigenvalues* are the same, $\lambda_i = \lambda$ for $1 \leq 1 \leq d - m^*$, and (22)-(23) reduce to

$$\frac{d}{dt} \log \lambda = (2m^* + 4)(1 - \lambda) - 2(d - m^*)\epsilon, \tag{B.1}$$

$$\frac{d}{dt} \log \epsilon = 2m^*(1 - \lambda) - 2(2 + d - m^*)\epsilon. \tag{B.2}$$

Those equations are an instance of *Lotka-Volterra equations* that have a long history for modeling competing species in ecology [5, 6]—here, the informative $\lambda$ and noninformative $\epsilon$ eigenvalues play the role of these species. (B.1)-(B.2) have three fixed points in the $(\lambda, \epsilon)$ space: the unstable solutions $(0, 0)$ and $(0, m^*/(2 + d - m^*))$, and the stable solution $(1, 0)$. The phase portrait of these equation is shown in Fig. B.6.

Let us analyze (B.1)-(B.2) when $d - m^* \gg 1$. In that case the dynamics of $\lambda$ and $\epsilon$ has two regimes: Initially the second term at the right hand side of these equation is the dominant term; since this term is negative, it means that both $\lambda$ and $\epsilon$ decrease from their initial values $(\lambda(0), \epsilon(0)) = (1, 1)$. In the second regime, $\epsilon$ becomes small enough that the right hand side of (B.1) becomes positive allowing $\lambda$ to bounce back up and grow towards its asymptotic value $\lim_{t \to \infty} \lambda(t) = \lambda^* = 1$ whereas $\epsilon$ continues to decreases so that $\lim_{t \to \infty} \epsilon(t) = 0$ converges to zero with a linear convergence rate. If we neglect the first term at the right hand side of (B.2), this equation can be solved exactly:

$$\epsilon(t) \approx \frac{1}{1 + 2(2 + d - m^*)t} \tag{B.3}$$

It turns out that this approximation is accurate in both regimes, because the first term at the right hand side of (B.2) is always sub-dominant. In the first regime, (B.1)-(B.2) implies that $\lambda(t) \approx \epsilon(t)$, and this goes on until the right hand side of (B.1) changes sign, indicating the start of the second regime. This occurs at time

$$t_0 \approx \frac{d - m^*}{2(2 + m^*)(2 + d - m^*)} = O(1), \tag{B.4}$$

similarly to the random Gaussian case discussed in the main text. Observe that at that time, we have $\lambda(t_0) = \epsilon(t_0) = (m^* + 2)/(d + 2)$, and passed that time $\lambda(t)$ starts to increase again, while $\epsilon(t)$ keeps decreasing. Therefore in this second regime we can neglect the last term at the right hand of (B.1) and solve this equation with the initial condition $\lambda(t_0) = \lambda_0 = (m^* + 2)/(d + 2)$. This gives the logistic growth

$$\lambda(t) \approx \frac{\lambda_0 e^{2(m^* + 2)(t - t_0)}}{\lambda_0 (e^{2(m^* + 2)(t - t_0)} - 1) + 1}. \tag{B.5}$$

From this equation, the time for $\lambda$ to reach its target $\lambda^* = 1$ is approximately

$$t_J \approx \frac{1}{m^* + 2} \log \frac{d + 2}{2(m^* + 2)}. \tag{B.6}$$

These approximations are remarkably accurate as we can observe in Fig. B.7, where we evaluate numerically the dynamics on the population loss (8) and compare the result with the approximation for $d = 512$, $m^* = 1$ and $n = 2048$. The left panel shows the evolution of the eigenvalues and the right one the generalization loss. The dotted line on the left is (B.3) and on the right is (27) shown in the main text.

Figure B.7: Evolution of the eigenvalues in the population loss (left) and generalization loss (right). Left panel: the solutions to (B.1)-(B.2) and the approximate solution (B.5) (dotted line). Right panel: exact loss from (21) compared to its approximation in (29) (dotted line) valid for small and large times. The vertical lines show the two times $t_0$ and $t_0 + t_J$.