[Reviews · NeurIPS 2020]

Review 1

Summary and Contributions: In this study, the authors investigate how generalization in one-hidden layer neural networks based on a teacher-student scenario affects the number of data samples n, the dimension of the samples d, and the dimension of the teacher's hidden layer m*. First, they show that, when m*=1, n/d>=2 is the condition for empirical error minimization to lead to generalization. They then show by a heuristic extension that the sample complexity threshold a_c depends on m*<d. They also investigate that this applies when optimizing with gradient descent (GD), and confirm it experimentally. Furthermore, they investigate the convergence rate of population loss in GD from the evolution of the eigenvalues of A, and confirm both theoretically and experimentally that the decay of convergence becomes exponential when n/d>a_c and m*>=d.

Strengths: The authors derive various conditions for generalization based on a teacher-student scenario. In particular, it is novel to show that, when m*< d, the sample complexity threshold depends on m* and that the decay of population loss in GD depends on the magnitude relation between m* and d. In addition, it is also a great contribution that these theoretically obtained results are confirmed experimentally.

Weaknesses: Although this paper is well written, there are several things that I don't understand. - What does "P", which appears in Eq. 3, lines 158, 194, etc., mean? - In the caption of Figure 1, the solid line is described as being predicted by Eq. 13. But isn't this estimated from Eq. 12 for m*<d and n=d(d+1)/2 for m*>=d? - Isn't "when n/d>=2" in line 199 a mistake for "when n/d<2"? - What is "Qm∗" in line 268? In addition, this paper seems to have ended abruptly because there is no summary. Although there is a limit to the number of pages, authors should discuss the limitations and applications of this work and future work.

Correctness: I'm not a specialist in this field, so I haven't been able tocheck the proofs in detail, but I think the discussion in this paper is correct.

Clarity: This paper is well organized, although there are some unclear points as described above.

Relation to Prior Work: The relationship between this study and previous ones is clearly written.

Reproducibility: Yes

Additional Feedback:


Review 2

Summary and Contributions: This paper studies learning with an over-parameterized two-layer neural network with quadratic activation where the input is standard Gaussian and the label is generated according to a planted two-layer neural network with quadratic activation. This paper obtains several results in this setting. a) if the planted model has only one neuron, then there is a phase transition of the uniqueness of the global minima in terms of the ratio between the number of data and the input dimension. And a heuristic extension of this result. b) A fine-grained analysis for the case of identity initialization together with a heuristic argument for the convergence rates for two regimes. c) Some empirical geometry insights obtained by the string method.

Strengths: The string method seems to be novel in the literature.

Weaknesses: The paper lacks of novelty and substantial results. The phase transition result in Theorem 3.1 is interesting. However, this only applies to the rank-$1$ phase retrival setting. The more interesting setting (and more related to neural networks) is when $m* \gg 1$. Unfortunetly, this paper only provides a heuristic formula. Theorem 4.2 only applies to the identity initialization, which essentially reduces the problem to several one-dimension dynamics analysis. The discussions on the convergence rates for $m^* \ll d$ and $m* \ge d \gg 1$ is interesting. However, there is no formal proof. I would like to recommend acceptance if this paper provides formal analysis to random initialization.

Correctness: Yes.

Clarity: Yes.

Relation to Prior Work: It would better if this paper provides a more explicient exposition on the improvement over existing results.

Reproducibility: Yes

Additional Feedback:


Review 3

Summary and Contributions: This paper focuses on the dynamics of optimization and the generalization properties of one-hidden layer neural networks with quadratic activation function in the context of a teacher-student set-up. 1. This paper shows a threshold value such that when the data size n is above the threshold, there is a finite probability that all spurious minim disappears in the limit as data dimension d increasing. When n is below the threshold, there exist spurious minima with probability 1 in the limit. 2. It evaluates the convergence rate of gradient descent in the limit of a large number of samples.

Strengths: 1. It studies the gradient descent flow on the empirical loss starting from random initialization and shows that it converges to a network that is a minimizer of the empirical loss. 2. Moreover, it evaluates gradient descent's convergence rate in the limit of a large number of samples.

Weaknesses: 1. First of all, the considered problem in this paper is quite over-idealized, e.g., one hidden layer NN rather than general DNN, GD flow rather than discrete GD, Gaussian input data rather than given general data, etc. It is known that there exists a gap between the gradient flow and its discretization. Although, these are the prevalent setting in the past few years, recently, there are abundant theoretical results for general DNN with general input. It makes less sense to concern on the old settings. 2. One of the main contributions to provide the convergence speed of GD on the case, m^*<d, seems too rough. More details need for the phase that other terms in Eq. (17) take over rather than the quadratic term. There is no Sec.C in the appendix. 3. The main results Theorem 3.1 only holds for a special case, i.e., m^* = 1. From the perspective of matrix analysis, it seems impossible to conclude the results rigorously without using the information from A^*.

Correctness: Almost right.

Clarity: Complete and clear.

Relation to Prior Work: Yes. However, It seems to have considered only a little more general case than Ref. [24].

Reproducibility: Yes

Additional Feedback: see weakness. ---------------------------------------------------------------------------- After reading the feedback, I found R2 and I almost focus on the same problems. For random initialization, I also believe that it still needs a lot of effort. The upper bound of E(A(t)) is clearly dependent on the condition number of A(0) instead of simply dividing the cases into full-rank and rank-deficient. Moreover, rather than only focusing on the full-rank case, the author may consider the problem uniformly and continuously, e.g., the MP-law from RMT may help to provide an asymptotic analysis for the random initialization since the universal distribution for the eigenvalues are given. Also, there may exist the non-asymptotic version, but more perturbation bounds are needed. BTW, due to my research background, I neglected the development of shallow neural networks with random Gaussian input. I am sorry about that and raise my score. However, considering this paper may need a fair amount of work to fix the existing problems, I do not recommend a strong accept.


Review 4

Summary and Contributions: This paper studies the learning dynamics of two-layer networks with quadratic activations in the overparametrized student-teacher setting. The paper describes conditions under which the set of global minimizers of the empirical loss can be expected to have good generalization (i.e. converge to the singleton set containing the teacher solution). The paper also analyzes the dynamics of gradient descent, showing they will converge to a minimizer of the empirical loss (under mild assumptions on the training data distribution) with a bound on the population loss.

Strengths: The paper clearly outlines its main contributions and reviews the relevant literature. The arguments behind some of the main theorems (e.g. Theorem 3.1) are elegant and concise. The paper validates its theoretical findings via extensive experiments. The experiments are well laid out and clearly motivated.

Weaknesses: The setting of the paper is limited in scope (i.e. shallow nets with quadratic activations and fixed second layer, teacher net with one hidden unit) and it's unclear what if any insights the theoretical arguments and/or experimental findings in the paper provide into more general architectures. Granted however, this is not necessarily a significant weakness; nets with quadratic activations are still in and of themselves an active area of interest for the ML community. More pressingly, some explanation is lacking at various points in the paper. Lemmas 2.1 and 2.2 should explicitly be derived in the Appendix, no matter how straightforward the derivation may be. Theorem 4.1 also needs a rigorous proof. For example, the authors should explicitly explain why "the only possibility for [gradient descent] not to do so would be to reach a critical point of Morse index 1 or above, and the probability of that event is zero from random initial data". Also, why must A(0) have full rank?

Correctness: As mentioned above, some of the theoretical arguments need further explanation. In line 33 of the Appendix, couldn't one of the \lamda_i be negative? In line 74 of the Appendix, how is the number of constraints imposed by requiring the eigenvalues of A_\perp to be non-negative. My understanding is that this is not an explicit calculation, but rather an extrapolation of the argument in lines 66-70 to the case where m* > 0. Could the authors clarify this point?

Clarity: For the most part, the paper is well written and motivated. Proof sketches of most of the major theorems are included in the main text. Experimental details are provided in the Appendix. Again however, some of the theoretical results should be explained more completely (see "Weaknesses" section). Also, even after reading Section B in the Appendix it is still somewhat hard to follow how the confidence intervals in Figure 1 are being calculated. What is the motivation behind the power law assumption?

Relation to Prior Work: Yes, no issues here.

Reproducibility: Yes

Additional Feedback: As of now I am recommending a score of 5 due to lingering issues regarding the completeness and clarity of some of the theoretical arguments and experiment descriptions. However, I expect these issues can be easily fixed by the authors and would be willing to change my score to an accept pending those changes (or of course a satisfactory rebuttal). EDIT (post author response): The authors did a fair job addressing the correctness/completeness concerns I outlined in my original review and (assuming the appropriate changes are made to the final version) I now feel the paper clears the bar for acceptance. That said, given the limited scope of the paper's setting (see the first part of the "Weaknesses" section in my original review), I think the results are perhaps not exciting enough to merit a strong accept. I've accordingly changed my score to a 6.

[Author Response · NeurIPS 2020]

We thank the reviewers for their careful consideration of our paper. Our analysis gives a new sample complexity threshold that shows how the generalization error in a teacher-student scenario depends on the number of teacher units $m^*$ and the input dimension $d$, a result highlighted by **R1**. We also derive the convergence rate of GD as a function of $m^*$ and $d$ in the limit of many samples, a result that both **R1** and **R3** commended. We are glad that the reviewers found our proofs elegant and concise (**R4**), and appreciated that we validate these theoretical findings by extensive numerical experiments that are well laid out and clearly motivated (**R1**,**R4**), and also use the string method as new tool in the ML literature (**R2**). We are encouraged that **R4** thought that our paper clearly outlines its main contributions and reviews the relevant literature. We are also grateful for the reviewers specific suggestions to improve our results. One primary concern was that some of the arguments were unclear or incomplete. We agree. *In the revised version we have followed the reviewers recommendations and added proofs and new results accordingly*, as detailed next. We also thank **R1** and **R3** for pointing out typos and residuals of previous notation, i.e. $P \leftarrow n$ and $Qm^* \leftarrow A^*$.

**R3**: *the considered problem in this paper is quite over-idealized.* We respectfully disagree. In particular, questions related to the generalization error or the convergence rate of GD remain mostly open even for shallow networks–here we give precise answers to these questions in the context of networks with quadratic activation functions, which, as **R4** remarks, "are still in and of themselves an active area of interest for the ML community."

**R3**: *[They] considered only a little more general case than Ref. [24].* We don't think this assessment is fair. In Ref. [24] the authors give a nice and thorough analysis of the case when $m^* > d$, i.e. $A^*$ is full rank and the threshold is at $n = d(d+1)/2$ by standard linear algebra results. The situation with $m^* < d$ that we focus on is relevant for phase retrieval and has a different sample complexity threshold that depends on $m^*$ and $d$, the proof of which requires analysis techniques from random geometry that are different from the ones used in Ref. [24].

**R1**: *authors should discuss the limitations and applications of this work and future work*; **R2**: *It would better if this paper provides a more explicit exposition on the improvement over existing results.* We will expand on these points in the introduction of our paper, in particular stress that most of the results in the literature focus on the empirical loss, showing that over-parameterization helps to find good minima (in particular refs.[8,15,16]). Here we focus on generalization error and convergence rate of GD, which are less studied and not as well understood.

**R2**: *Theorem 4.2 only applies to the identity initialization. [..] I would like to recommend acceptance if this paper provides formal analysis to random initialization.* **R3**: *More details need for the phase that other terms in Eq. (17) take over rather than the quadratic term.* Our analysis of Eq. (14) was kept informal for readability, but it is not hard to make it rigorous and we will add a proof of Eqs. (20) and (23) in the Appendix. Also, we had focused on the case of $A(0) = \text{Id}$ because, if the initial weights $w_i(0)$ are independent standard Gaussian vectors, $A(0) = m^{-1} \sum_{i=1}^{m} w_i^T(0) w_i(0)$ is close to the identity as long as $m \gg d$ by the Law of Large Numbers. However, the convergence analysis can be generalized to arbitrary initial $A(0)$ with full rank ($m \geq d$) and we can show that: (i) there exists a constant $C > 0$ such that, for all $t \geq 0$, $E(A(t)) \leq 1/(1 + Ct)$ (including when $m^* < d$, i.e. when $A^*$ is rank-deficient, which is the difficult case for analysis) and (ii) Eqs. (20) and (23) in the paper still hold as $t \to \infty$. We will add this result as a new theorem along with its proof based on the analysis of the dynamics of the off-diagonal term of $A(t)$: since $A(t) \to A^*$, these terms must eventually decay to zero at a rate that can be shown to be exponential, which is enough to establish the results (including the non-asymptotic convergence rate, using convexity of the loss).

**R3**: *The main results Theorem 3.1 only holds for a special case, i.e., $m^* = 1$. From the perspective of matrix analysis, it seems impossible to conclude the results rigorously without using the information from $A^*$.* The proof of Theorem 3.1 for $m^* = 1$ is based on a geometric rephrasing that uses the rank 1 structure of $A^*$. Extending this proof to $A^*$ with rank $m^* > 1$ seems nontrivial, we agree, as the geometrical problem becomes more complex. Still our numerical experiments strongly suggest that the result from our heuristic argument is correct.

**R4**: *Lemmas 2.1 and 2.2 should explicitly be derived in the Appendix. Theorem 4.1 also needs a rigorous proof.* We will add these proofs: Lemma 2.1 follows by direct calculation from Eqs. (3) and (4); Lemma 2.2. follows from Lemma 2.1 by averaging over the Gaussian weights using Wick's theorem; and Theorem/Proposition 4.1 follows from the Stable Manifold Theorem which states that the GD flow reaches a local minimum of the empirical loss with probability one, and all these minima have zero loss by convexity of the loss in $A$ – the result is stated under the assumption that $A(0)$ has full rank, for otherwise rank deficiency introduces a nontrivial constraint which may preclude the dynamics to reach the minimum; we note however that this assumption is sufficient but may not be necessary.

**R4**: *In line 33 of the Appendix, [..] In line 74 of the Appendix, [..]* When $m^* = 1$, $A^*$ has only one nonzero eigenvalue and at most one eigenvalue of $A - A^*$ is negative. When $m^* > 1$, the number $\frac{1}{2}(d - m^*)(d - m^* - 1)$ of constraints is obtained by requiring that $A^\perp$ be nonnegative definite in the $d - m^*$ subspace were it lives, which gives (A.13), but this argument is only heuristic for the reason stated in lines 78-80.

**R4**: *How the confidence intervals in Figure 1 are being calculated? What is the motivation behind the power law assumption?* The extrapolations consider a power law divergence at the transition [Cugliandolo, Kurchan'93] and the confidence interval comes assuming a t-student distribution of the samples.

[Meta-Review · NeurIPS 2020]

Reviews for this paper are mitigated, in particular some reviewers were concerned about some missing proofs. On the other hand, the paper studies an important problem and carries a nice analysis that integrates numerical experiments, heuristic derivations and rigorous proofs in a meaningful way; and the reader learns a lot about such models (quadratic 2-layer networks with sparse teacher). I thus suggest accept. However, some non-trivial results (Eq. 14, 20, 23,...) were stated and claimed to be true but without proof which is not acceptable. It is thus necessary that the authors spend a lot of effort writing the missing proofs thoroughly because it will not be possible to review those proofs again (and of course all the other changes proposed in the rebuttal should be implemented). Overall, for such a paper that contains true statements, conjectures and heuristics, it is very important to emphasize on the "truth status" of each statement, and "true statements" should have a proof.